# DPSC Products Accelerate Wound Healing in Diabetic Mice through Induction of SMAD Molecules

**DOI:** 10.3390/cells11152409

**Published:** 2022-08-04

**Authors:** Carl J. Greene, Sarah Anderson, Derek Barthels, Md Sariful Islam Howlader, Suman Kanji, Jaganmay Sarkar, Hiranmoy Das

**Affiliations:** Department of Pharmaceutical Sciences, Jerry H. Hodge School of Pharmacy, Texas Tech University Health Sciences Center, ARB Suite 2116, 1406 South Coulter Street, Amarillo, TX 79106, USA

**Keywords:** DPSC products, cutaneous wounds, healing, inflammation, NF-kB, Smad

## Abstract

Despite advances in diabetic wound care, many amputations are still needed each year due to their diabetic wounds, so a more effective therapy is warranted. Herein, we show that the dental pulp-derived stem cell (DPSC) products are effective in wound healing in diabetic NOD/SCID mice. Our results showed that the topical application of DPSC secretory products accelerated wound closure by inducing faster re-epithelialization, angiogenesis, and recellularization. In addition, the number of neutrophils producing myeloperoxidase, which mediates persisting inflammation, was also reduced. NFκB and its downstream effector molecules like IL-6 cause sustained pro-inflammatory activity and were reduced after the application of DPSC products in the experimental wounds. Moreover, the DPSC products also inhibited the activation of NFκB, and its translocation to the nucleus, by which it initiates the inflammation. Furthermore, the levels of TGF-β, and IL-10, potent anti-inflammatory molecules, were also increased after the addition of DPSC products. Mechanistically, we showed that this wound-healing process was mediated by the upregulation and activation of Smad 1 and 2 molecules. In sum, we have defined the cellular and molecular mechanisms by which DPSC products accelerated diabetic wound closure, which can be used to treat diabetic wounds in the near future.

## 1. Introduction

Diabetes continues to affect millions of people around the globe and is associated with several secondary complications, including peripheral neuropathy, peripheral artery disease, and deformity of soft tissue and bone [1]. Diabetic foot ulcers are one of the most severe conditions found in diabetics that affect as many as 9.1 to 26 million diabetics worldwide every year [2]. Several factors, such as peripheral neuropathy, peripheral vascular disease, and trauma contribute to the development of foot ulcerations; however, the main cause is considered to be impaired wound healing that leads to severe pathogenesis [3,4]. The management of chronic wounds caused by diabetes is a major cause of hospitalization and injury that costs approximately $25 billion in the United States [5].

Wound healing is a dynamic process, which initiates as soon as the integrity of the tissue is disrupted. The process of healing has been distributed into four major parts, namely hemostasis, inflammation, proliferation, and remodeling. Activated platelets are important in the first stage of tissue repair by encouraging the adhesion of neutrophils, induction of several growth factors such as transforming growth factor-β (TGF-β), and induction of monocyte migration to the wound bed [6]. The neutrophils play one of the most important roles in the stewardship of infection by producing a wide range of cytokines and growth factors, helping in the differentiation of circulatory monocytes into activated or pro-inflammatory (M1) and anti-inflammatory (M2) subsets [7]. This aids the activation of fibroblasts, angiogenesis of new capillaries, and remodeling of capillaries throughout the proliferative phase [7].

The diabetic condition consists of a chronic low-grade inflammatory phase characterized by an increased ratio of M1/M2 and prolonged inflammatory responses via increased levels of inflammatory cytokines such as interleukin-1 (IL-1), interleukin-6 (IL-6), and tumor necrosis factor-α (TNF-α), in a long run impairs healing activities after injuries [8]. Literature shows that there is an inverse relationship between neutrophil activities and effective wound healing, and in diabetes, an elevated level of neutrophils is found that damages the wound-healing process [9]. The diabetic wound is also associated with impaired angiogenesis and endothelial progenitor cell dysfunction, as well as the disruption of ECM secretion [10].

Current advancement for diabetic wound control and therapy only offers limited relief and does not address the underlying cause of the condition, which results in diabetic wounds that never heal or heal very slowly [5]. There is considerable interest in the possibility of utilizing stem cell therapy for the treatment of cutaneous wounds, including in diabetic conditions [11,12,13,14]. An important effect of stem cells transplanted into a wound is their release of cytokines and growth factors capable of promoting cell proliferation, angiogenesis, ECM remodeling, and immunomodulation [15]. In recent years, induced pluripotent stem cells (iPSC) have been recognized as a novel therapeutic option with the advantage of being able to be used in animal models of wound healing [16]. Dental pulp stem cells (DPSC) provide access to stem cell reservoirs more efficiently since they are isolated from third molars, which are generally discarded as medical waste when they are no longer needed. Moreover, they have the advantage of being extremely easy to isolate, maintain as cultures, and preserve for long periods without losing their stem cell characteristics. Their anti-inflammatory properties, as well as their potential to furnish essential growth factors that would represent a contribution to improving wound healing, are well documented. However, the exact mechanisms of these actions are not yet clear. The secretory products of DPSC consist of numerous growth factors that contribute to the wound-healing process. Our previous studies [17,18,19] indicate that the DPSC could be an attractive contour for the treatments of diabetic wounds as well as for wound healing. In this study, we investigated whether we could treat wounds effectively in diabetic NOD/SCID mice using products collected from DPSC, and defined the mechanisms by which these stem cell secretory products work.

## 2. Materials and Methods

### 2.1. Reagents and Antibodies

An MRC-5 cell line was obtained from the American Type Culture Collection (ATCC, Manassas, VA, USA, #TIB-71); PVDF membrane (Bio-Rad Laboratories, Hercules, CA, USA, #1620115); enhanced chemiluminescence (ECL) (Amersham Pharmacia Biotechnology, Amersham, UK, #RPN2232). DMEM high-glucose media was purchased from GIBCO (Waltham, MA, USA, #11-965-092). The High-Capacity RNA-to-cDNA Kit (#4387406) and SYBR Green PCR Kit (#4309155) were the products of Applied Biosystems (Waltham, MA, USA). Additional materials included myeloperoxidase antibody (purchased from Abcam, Cambridge, UK, #ab9535), a-SMA Ab (purchased from Sigma-Aldrich, St. Louis, MO, USA, #A2547), collagen 1 (purchased from Santa Cruz Biotechnology, Dallas, TX, USA, #sc-8784). The NFkB (pP65 #3033S, p65 #8242S), pSMAD2 (3108T), SMAD2 (#5339T), pSMAD3 (#9520T), SMAD3 (9523T), p38MAPK (#4511S), p38MAPK (#9212S), pJNK (#9251) and GAPDH (#2118S) antibodies were purchased from Cell Signaling Technology, Danvers, MA, USA.

### 2.2. Dental Pulp-Derived Stem Cells Isolation and Expansion

Isolation and expansion of DPSC were carried out following our previously published method [17]. In brief, human third molars were collected from dental clinics with the consent of donors. Molars were washed three times with phosphate-buffered saline (PBS) containing 1% penicillin/streptomycin/glutamine (PSG) (Gibco, Thermo Fisher, Waltham, MA, USA). The molar was split in half, and the dental pulp was extracted from the interior cavity. The extracted pulp was then minced into 1 mm cubes and plated on 60 mm culture dishes using α-modified eagle medium (MEM) (Gibco) consisting of 20% fetal bovine serum (Hyclone, Thermo Fisher, Logan, UT, USA) and 1% PSG. Cells growing out of the pulp tissues were scraped and plated on 100 mm dishes and considered as passage 1. Experiments were carried out using these cells and supernatants were collected from passages 4 to 10.

### 2.3. Scratch Wound Assay

The human fibroblast cell line MRC5 was used for the scratch wound-healing assay. MRC5 cells were cultured in the wells of a 6-well plate, and the next day, after making a scratch, cells were cocultured with DPSC in transwell inserts at a 1:100 DPSC: MRC5 for 12 and 24 h periods in the presence of DMEM medium containing high glucose (4500 mg/L), 1% FBS, and 1% penicillin and streptomycin. This medium configuration was used for both the control and treatment condition. Images were taken using a bright-field microscope. Cell proliferation was measured using the MRI wound-healing macro tool for ImageJ software (Version 1.53s, NIH, Washington, D.C., USA). The MRI wound-healing tool measured the area of the wound. The formula below was used to measure wound closure, where *Area_to_* represents the initial wound area after the scratch and *Area_t_* represents the area of the wound at the time when subsequent data were collected.
(1)$$\frac{Area_{to}-Area_{t}}{Area_{to}}\times 100\%$$

### 2.4. Generation of Diabetes in NOD/SCID Mice

All murine experiments were conducted with prior Institutional Animal Care and Use Committee approval of TTUHSC, Lubbock, TX. Twenty immunocompromised NOD/SCID mice were purchased from Jackson Laboratory (Bar Harbor, ME, USA). Mice 8 to 10 weeks old were used to avoid any hormonal influences in the healing process. After acclimatization, mice fasted for four hours before injection with streptozotocin (STZ). A 50 mg/kg dose of STZ (Teva Parenterals Inc., Irvine, CA, USA) was dissolved in citrate buffer (pH 4.2) and injected intraperitoneally once each day for 5 consecutive days. To assess the development of diabetes, non-fasted blood glucose was measured once a week for six weeks using the AlphaTRAK blood glucose monitoring system (Abbott Laboratories, North Chicago, IL, USA). Blood glucose was measured before injection to establish a baseline level. The mice were anesthetized, and the tip of the tail was nicked to obtain a drop of blood. A blood glucose level greater than 300 mg/dL was considered diabetic in this mouse model.

### 2.5. Generation of Full-Thickness Excisional Cutaneous Wound and Application of DPSC Supernatant

DPSC were cultured in α-minimum essential medium (α-MEM) until 80% confluency. The media was removed, and the cells were washed with PBS. Fresh PBS was then added and collected after 12 h for in vivo experiments. This is hereafter referred to as the DPSC product. After four weeks of STZ injection, a cutaneous wound was generated using a full-thickness 8 mm punch on the dorsal area of the mouse. Two hours after cutaneous excision, the DPSC-derived products were sprayed onto the wound using a cosmetic airbrush twice per day until animals were sacrificed. PBS alone was sprayed on the wounds of the animals used as vehicle controls. On days 5, 10, and 15, six mice were sacrificed for wound tissue harvest (20 mice/group). Skin samples included the wound and 2 mm of the surrounding skin.

### 2.6. Immunohistochemistry

Immunohistochemistry was performed as previously described [14]. Skin tissues were isolated after mice were sacrificed. Tissues were fixed in 10% formalin-PBS buffer. They were then embedded in a paraffin block and sectioned into 4 µm segments. Sections were then deparaffinized and stained with hematoxylin and eosin (H&E), Masson’s trichrome, and myeloperoxidase. Antigens were retrieved by baking slides in a microwave for 5 min in citrate buffer at a pH of 6.0. Samples were treated to prevent non-specific binding and stained using VECTASTAIN Elite ABC kits, according to protocols provided by the manufacturer (Vector Laboratories Inc, Burlingame, CA, USA). This was performed using primary antibodies for myeloperoxidase (MPO). These were detected using 3,3′-Diaminobenzidine (DAB). Stains were visualized with an Olympus ix81 microscope (Olympus, Tokyo, Japan) using a 40x objective, and images were analyzed using Slidebook 5.0 × 64 software (Version 5.0.0.34, Intelligent-Imaging, Edmonton, Canada). Image analyses of specific staining for neutrophils and collagen depositions were performed by using Fiji in ImageJ software. A star (*) indicates the statistical significance (*p* < 0.001) between the control vs. the DPSC^℗^-treated group.

### 2.7. Multiplex ELISA

DPSC were cultured in PBS in a 24-well plate at a density of 5 × 10^5^ cells/well for 3 h and 24 h. PBS with no DPSC culture was used for the control. Collected supernatants were mailed to Ray Biotech, Atlanta, USA, for high-throughput analysis using multiplex ELISA. Each of the samples was in duplicate and analysis was performed four times. Statistical analysis was performed with the raw data and the standard error of means was provided with the graph.

### 2.8. Primary Murine Fibroblast Cell Isolation and Culture

A primary murine dermal fibroblast cell line was established from skin punch biopsies from C57BL/6 mice. Primary murine dermal fibroblast cells were maintained in DMEM supplemented with 10% FBS and grown in 5% CO_2_ at 37 °C. Cells were used from passage 4 to passage 10 for the experiments.

### 2.9. Quantitative RT-PCR Analysis

Total RNA was isolated from cells after various experiments, as mentioned. The reverse-transcription of 1 µg of mRNA was performed using High-Capacity RNA-to-cDNA Kit (Applied Biosystems, Waltham, MA, USA). Reactions were conducted in SYBR Green master mix (Applied Biosystems, Waltham, MA, USA) using a CFX96 Real-Time System (Bio-Rad Laboratories, Hercules, USA) with Bio-Rad CFX Manager software (Version 3.1, Bio-Rad Laboratories, Hercules, CA, USA). The primers used were as follows: Mouse (m) *IL-6*, m *Transforming Growth Factor-β1* (*TGF-β1*), m *IL-10*, m *Fibronectin*, m *α-smooth muscle actin* (*α-SMA*), and m *collagen 1A1* (*Col1A1*). The results were expressed as 2^−ΔΔCT^, yielding fold changes in expression compared to the controls. Following are the mouse primer sequences used for the PCR analysis. *IL-6*: Forward- 5′ ATC CAG TTG CCT TCT TGG GAC TGA 3′, Reverse- 5′ TAA GCC TCC GAC TTG TGA AGT GGT 3′; *β-actin*: Forward-5′ AAT GTG GCT GAG GAC TTT GT 3′, Reverse-5′ GGG ACAT TCC TGT AAC CAC TTA TT 3′; *IL-10*: Forward- 5′-TAGAGCTGCGGACTGCCTTCA, Reverse-5′-ATGCTCCTTGATTTCTGGGCCAT; *Col1A1*: Forward-5′ GTTGGAAACGGAGTTGCATAAG 3′, Reverse-5′ CAGAGTCTGGGTGAGTGTTAAG 3′; *Fibronectin*: Forward- 5′ TACGGAGAGACAGGAGGAAATA 3′, Reverse 5′ CATACAGGGTGATGGTGTAGTC 3′; *α-smooth muscle actin* (*α-SMA*): Forward 5′ GACTCTCTTCCAGCCATCTTTC 3′, Reverse 5′ GACAGGACGTTGTTAGCATAGA 3′; *Transforming Growth Factor-β1* (*TGF-β1*): Forward 5′ GGTGGTATACTGAGACACCTTG 3′, Reverse 5′ CCCAAGGAAAGGTAGGTGATAG 3′.

### 2.10. Western Blot Analysis

Total protein was isolated from murine fibroblasts for Western blot (WB) using RIPA buffer. Briefly, cells were washed 3 times with ice-cold 1x PBS and lysed using the pre-cooled RIPA lysis buffer on ice for protein extraction. Pellets were removed after centrifugation. The protein concentrations were measured using the BCA protein assay kit, according to the manufacturer’s protocol. Equal amounts of proteins (40 μg) were separated by SDS-PAGE gel electrophoresis and transferred to the PVDF membrane. Then, membranes were blocked with 5% non-fat milk for 1 h at room temperature and incubated with primary antibodies overnight at 4 °C. After that, membranes were incubated with appropriate HRP-conjugated secondary antibodies for 2 h at room temperature. Protein bands were visualized by ECL.

### 2.11. Application of DPSC Supernatant to Dermal Fibroblast

DPSC were cultured in α-minimum essential medium (α-MEM) until 80% confluency. The media was removed, and the cells were washed with PBS. Fresh MEM containing 1% FBS was then added and collected after 12 h for in vitro experiments. Four hours after seeding dermal fibroblasts, the media was removed and the DPSC supernatant was added. Twenty-four hours after seeding the fibroblasts, 10 ng/mL of TNF-α were added to the cell culture. Protein was collected 30 min and 24 h later.

### 2.12. Immunocytochemistry

Cultured cells were fixed with 4% PFA for 30 min and washed with 1x PBS; cells were then permeabilized with 0.1% Triton X-100 for 15 min at room temperature, and blocked with 1% BSA for 30 min. Then, cells were incubated with 200 μL of primary antibody (1:200 dilution) overnight at 4 °C. The next day, cells were washed with 1x PBS, followed by incubation with 200 μL of secondary antibodies (Alexa Fluor 488, #A11001 and Texas red, #PA1-28662; 1:2000 dilution; Invitrogen Corporation, Waltham, MA, USA) for 45 min in the dark. After incubation, cells were washed thrice with 1x PBS and mounted using DAPI on glass slides. Fluorescence images were captured using an Olympus ix81 microscope using a 40x objective, and images were analyzed using Slidebook 5.0 × 64 software (Version 5.0.0.34, Intelligent-Imaging, Edmonton, Canada).

### 2.13. Statistical Analysis

All in vitro experiments were carried out in triplicate. Quantitative data are represented as mean ± SEM unless otherwise stated. We used a 2-tailed Student’s *t*-test to analyze differences between groups. The *p*-value of 0.05 was chosen as the threshold of statistical significance.

## 3. Results

### 3.1. Effect of DPSC Products on Wound Closure in Diabetic NOD/SCID Mice

Streptozotocin (STZ)-induced diabetes was developed in NOD/SCID mice and was confirmed by measuring blood glucose levels (Figure 1A). In the absence of STZ injection, non-fasting blood glucose levels in control mice were 188.2 mg/dL (Figure 1A). This value reached 718 mg/dL on day 21 and 744 mg/dL on day 42 after STZ injection, indicating induction of diabetes in NOD/SCID mice and maintenance of hyperglycemia throughout the experimental period (Figure 1A).

The wounds generated in diabetic mice were treated topically with DPSC products by spraying directly on the wound and showed accelerated healing compared to the control group, which acted as a PBS vehicle control. Treatment groups showed a notable increase in wound closure by day 9, and wounds were closed or almost completely closed after 15 days of treatment, while wounds in control mice were still open on day 11 (68/55% wound closure), day 13 (73/64% wound closure), day 15 (85/74% wound closure) (Figure 1B,C).

### 3.2. The Effects of DPSC on Cutaneous Wound Cellularization

When wound sections were examined histologically, animals treated with DPSC products showed a remarkable formation of new granulation in wound tissues throughout the treatments compared to the untreated animals. In addition, DPSC products-treated animals displayed epithelialization on day 15, as evidenced by increased collagen (CG) deposition, whereas diabetic wounds in control animals showed only the beginnings of epithelialization. Animals receiving DPSC products also exhibited increased collagen deposition and fibroblast recruitment. This was an improvement over the diabetic control wounds, which showed only the beginnings of epithelialization on day 15 with very little recruitment of collagen or fibroblasts, indicating prolonged inflammation along with the remaining blood vessels (Figure 2A).

Additionally, we used an immunohistochemical analysis of wound tissue sections to validate neutrophil infiltration by analyzing the specific marker myeloperoxidase. On day 10, immunohistochemical analysis of the wound bed showed that the number of myeloperoxidase-positive neutrophils was noticeably reduced compared to the controls, which indicated that the DPSC products regulated the infiltration of neutrophils into the wound bed (Figure 2B).

### 3.3. Effect of DPSC Products on Collagen Synthesis in the Wound Tissues

Masson’s trichrome staining was used to investigate the effect of DPSC products in the formation of collagen in the wound tissues to restore the wound-healing process. The use of DPSC products resulted in greater collagen deposition in diabetic wound tissues, as seen by the blue staining on days 10 and 15, than that found in wounds of control diabetic animals, indicating that collagen production was restored (Figure 3).

### 3.4. Effect of DPSC Products on the Expression of Inflammatory and Remodeling Genes and Proteins in Primary Fibroblasts

A separate evaluation of primary murine fibroblasts in vitro indicates that the attenuation of inflammatory factors such as *IL-6*, as well as upregulation in wound remodeling genes, may have contributed to the increased rates of wound closure. Diabetic wounds treated with DPSC products showed a significant reduction in pro-inflammatory activity indicated by a decreased level in neutrophil infiltration, a decrease in proinflammatory cytokine production, and a simultaneously increased expression of anti-inflammatory markers such as *TGF-β1* and *IL-10* in fibroblasts in the presence of DPSC products (Figure 4A). Moreover, the addition of DPSC products to the fibroblasts in an inflammatory environment also significantly increased the expression of *α-SMA*, *fibronectin*, and *Col1A1* genes, all of which are essential for the wound regeneration process.

To evaluate the effect of DPSC in the inflammatory condition, we have cultured fibroblast cells in the presence of TNF-α. Immunocytochemistry was performed utilizing the fibroblast marker α-SMA and collagen 1. A higher level of the a-SMA and collagen 1 molecules was detected when DPSC were present during inflammatory conditions induced by TNF-α compared to control cells (Figure 4B).

### 3.5. Effect of DPSC Products on the Experimental Wound Healing

To determine the effect of the DPSC products on wound healing, a scratch assay was performed using MRC5 human fibroblast cells. Our results showed that the fibroblast cells have a higher rate of wound closure at 12 h (45% vs. 15% wound closure) and at 24 h (60% vs. 25% wound closure) when co-cultured with DPSC, compared to fibroblasts cultured without DPSC (Figure 5A–C).

### 3.6. Identification of Factors Present in DPSC Products

To define what factor play critical roles in wound healing, we analyzed the secretory products of the DPSC. We found that DPSC secrete a variety of factors such as cytokines, growth factors, and angiogenic factors at various time points (Figure 6), which may be responsible for mediating faster wound healing in both in vivo and in vitro conditions.

### 3.7. Effect of DPSC Products in Translocation of NFkB and Level of Signaling Molecules

We have investigated the effect of DPSC products on a master regulator of inflammation, NFkB, and its translocation using primary fibroblasts in inflammatory conditions using the immunocytochemistry method. Our results showed that the NFkB translocation to the nucleus was remarkably reduced after coculture with DPSC in a dose-dependent manner (Figure 7A). We also investigated the activation of NFkB (pP65) using the Western blot method and found that the DPSC products decreased the level of activation of NFkB (pP65) in inflammatory conditions (Figure 7B). To investigate the molecular mechanism by which DPSC products mediated the wound healing, we found that the activation of SMAD2 and SMAD3 was highly increased; however, the level was not remarkably altered for p38MAPK and JNK molecules (Figure 7B).

## 4. Discussion

The management of diabetic wounds is an area with numerous advanced therapeutic options available for patients; however, many patients still undergo amputations as a result of failed therapies, making it clear that alternative therapies must be developed that have greater efficacy [20]. In recent years, some pilot studies have been carried out with the hopes of treating diabetic wounds with stem cells taken from the patient’s own body. However, these studies have only been partially successful [21]. A major characteristic of DPSC is their ability to behave as mesenchymal stem cells (MSCs), which includes their ability to induce angiogenesis [22]. Research has indicated that DPSC demonstrate significant potential for treating disease conditions such as diabetes and impaired wound healing [23]. Diabetes is a well-established disease model, and chemical induction of diabetes is a simple, reliable, and inexpensive method for creating the disease in rodents, which is also suitable for use in clinical trials [24,25]. In NOD/SCID mice that are immunocompromised, we have developed a stable diabetic condition using STZ injections. This method provided a platform for the integration of human cells without rejection by the immune system.

Diabetes causes slower healing of wounds or a slower rate of re-epithelialization, which would result in a lesser role for angiogenesis and recellularization in the healing process (Figure 1A–D) [11,12]. Cellularization is achieved during the inflammatory phase of wound healing through the recruitment of neutrophils, macrophages, and fibroblasts that contribute to the release of cytokines such as interleukins (IL), TNF- α, and growth factors [26]. A greater amount of cellularization was mediated within the injured tissues via the recruitment of fibroblasts, which mediates the deposition of collagen (Figure 2). Neutrophils are major inflammatory regulators, and they are considered among the key players in the diabetic wound-healing process. It is necessary, therefore, that the inflammation is present during the initial stages to start the healing process, but persistent inflammation that remains during this healing process constitutes the most significant cause of failure in wound healing. Neutrophils promote inflammation by infiltrating the wound on the wound surface and secreting cytokines like TNF-α that trigger inflammatory signaling pathways [27,28]. During the later stages of wound healing (day 10), we observed a significant increase in the neutrophil population in the wound bed of diabetic NOD/SCID mice, whereas the number of neutrophils significantly declined following the application of DPSC products (Figure 2B). A pro-inflammatory response at the diabetic wound bed was further demonstrated by higher levels of TNF-α in the tissues of control diabetic animals as compared to DPSC products-treated diabetic animals, which perfectly correlated with improved wound healing in diabetic mice after the application of DPSC products. Another important cytokine is IL-10, which is considered one of the most important anti-inflammatory cytokines in the healing process after an injury [29]. In the present study, diabetic NOD/SCID mice showed a significant reduction in wound inflammation following the application of the DPSC products. This may partly be attributable to the presence of significantly higher levels of the anti-inflammatory genes *IL-10* and *TGF-β*, and the decreased level of the pro-inflammatory gene *IL-6* in the wounds (Figure 4). The chemokine monocyte chemoattractant protein-1 (MCP-1) is responsible for recruiting monocytes to inflammatory sites [30]. Granulocyte-macrophage colony-stimulating factor (GM-CSF) is an immunoregulatory cytokine that is involved in the promotion of immune tolerance [31]; our findings also support this hypothesis (Figure 6).

Fibronectin is an adhesive molecule that is profoundly important for the healing of wounds, particularly in extracellular matrix (ECM) formation and in re-epithelialization [32]. During diabetic conditions, fibronectin levels were low, and those levels were restored after the application of the DPSC products. In general, diabetes results in impaired wound contraction as a result of the impaired proliferation of dermal fibroblasts and reduced expression of α-SMA on these fibroblasts (Figure 3). In diabetic wounds, increased levels of TNF-α can reduce healing effectiveness, since this pro-TNF-α suppresses the expression of α-SMA in human dermal fibroblasts [33]. Our study has shown that the genetic expression of *α-SMA* is significantly reduced under diabetic conditions but was restored after the application of DPSC products (Figure 3). Skin fibroblasts also secrete collagen proteins, which are significant components of the ECM. The *collagen type 1* gene (*col1A1*) replaces fibrin clots during wound healing [34]. In our experimental wound model, we observed that the collagen was decreased during TNF-α-mediated inflammatory conditions that were effectively controlled with DPSC secretory products (Figure 5A). We also revealed a significantly increased level of collagen staining after the application of DPSC products to the diabetic wound in mice on days 10 and 15 (Figure 3). A similar result was also seen in the experimental wound model using primary fibroblasts, where collagen was significantly increased after the application of the DPSC products (Figure 5A).

TNF-α stimulates the signal for matrix metalloproteinases (MMPs), which are collagenolytic in chronic wounds and are activated through the NFκB pathway [35]. We found that the NFκB and TNF-α protein levels were significantly decreased in diabetic wounds after the application of the DPSC products, suggesting a DPSC products-mediated regulatory interaction through activation of NFκB. In this way, DPSC were able to reduce downstream inflammatory gene expressions by reducing the transcriptional activity mediated by NFκB, which is supported by our Western blot and ICC data (Figure 7A,B). Through signaling pathways analysis, we finally established that the DPSC products activated the SMAD signaling and not the other signaling pathways tested such as p38 MAPK, and JNK (Figure 7B). This is in agreement with previous literature, which shows that the TGF-β/Smad signaling pathway is involved in the process of tissue regeneration [36]. When Smad2/Smad3 are phosphorylated, they reactivate the Smad7 promoter, and Smad7 then inhibits TGF-β expression (negative feedback regulation) [37]. By regulating the TGF-β/Smad signaling pathway, DPSC products also promote cutaneous wound healing. In both our in vitro and our in vivo studies, it was shown that DPSC products inhibited the NFκB transcriptional activation pathway and downregulated the inflammatory milieu in diabetic conditions and resulted in accelerated wound healing.

## Figures and Tables

**Figure 1 cells-11-02409-f001:**
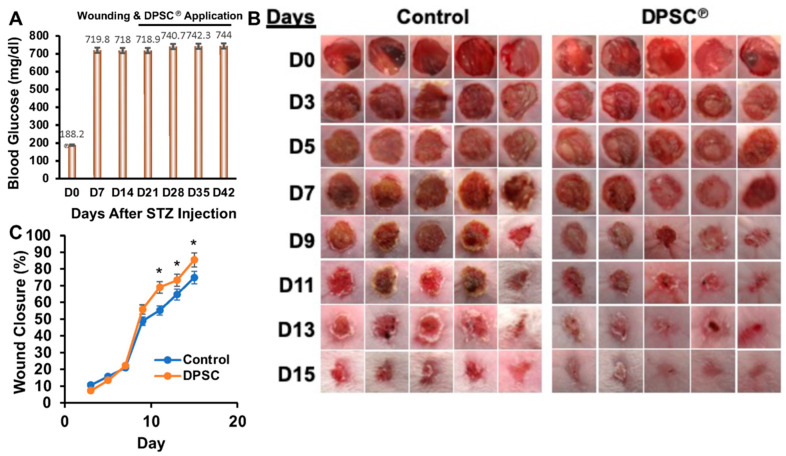
Induction of diabetes in NOD/SCID mice, generation of the cutaneous wound, and DPSC products therapy. (**A**) Non-fasting blood glucose level was measured on day 0 and once a week, the wound was generated on day 21 and DPSC products application (from day 21 to 42) for the whole of the study. (**B**) Wound images were captured every alternate day and presented. (**C**) Percentage of wound closures (±SEM) are shown graphically. A star (*) denotes a significant change (*p <* 0.05) compared to the control conditions.

**Figure 2 cells-11-02409-f002:**
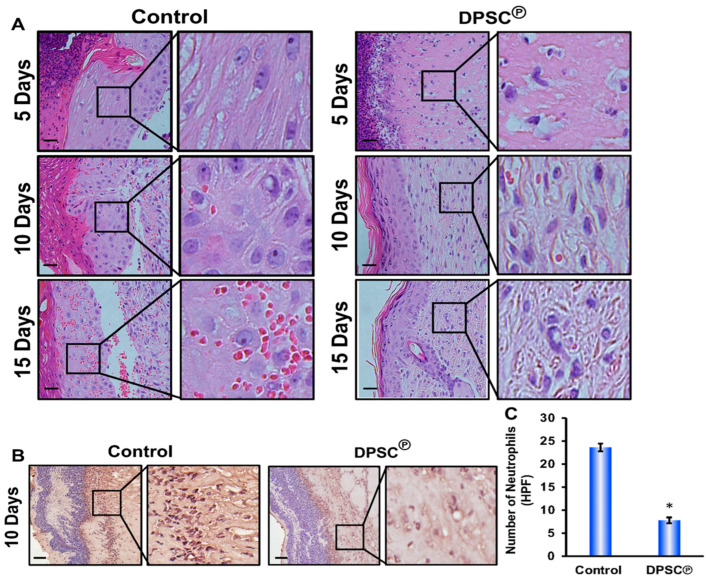
Increased cellularization and epithelialization in wounds after topical application of DPSC℗. (**A**) Images at different time points (days 5, 10, and 15) after hematoxylin and eosin staining. (**B**) Images captured on day 10 after myeloperoxidase staining. (**C**) Graphical presentation of evaluated neutrophil staining images performed by using Fiji in ImageJ software. A star (*) indicates the statistical significance (*p* < 0.001) between the control vs. the DPSC℗-treated group.

**Figure 3 cells-11-02409-f003:**
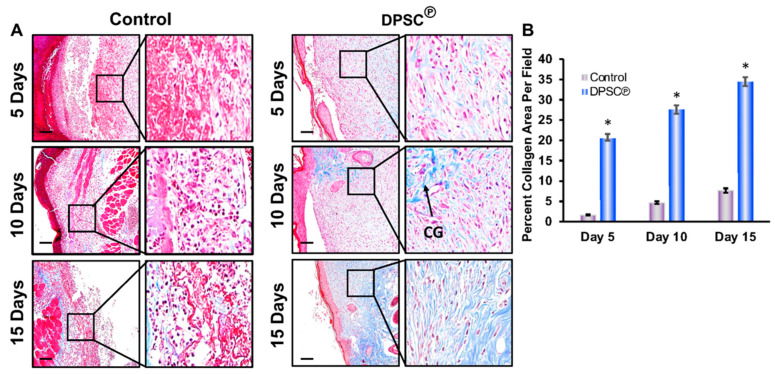
(**A**) Enhanced collagen deposition in wounds after topical application of DPSC℗. Trichrome staining was performed on the wound tissues and images were captured at different time points (days 5, 10, and 15). The arrowhead indicates collagen deposition (CG). (**B**) Graphical presentation of evaluated collagen staining images performed by using Fiji in ImageJ software. A star (*) indicates the statistical significance (*p* < 0.001) between the control vs. the DPSC℗-treated group.

**Figure 4 cells-11-02409-f004:**
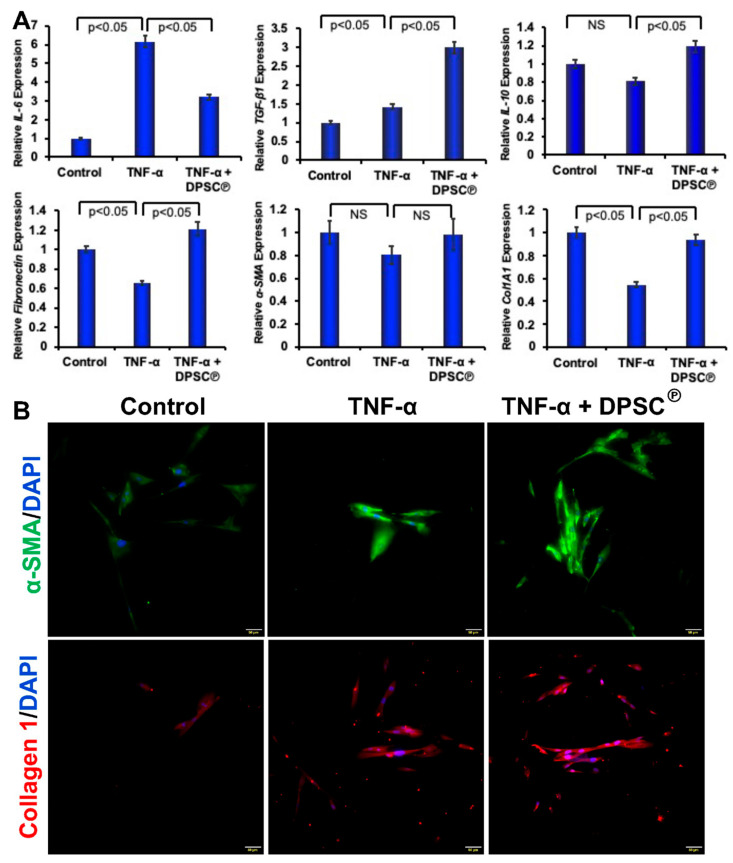
(**A**). The addition of DPSC℗ reduces inflammatory and increases anti-inflammatory genes in primary fibroblasts. Quantitative RT-PCR was performed in primary fibroblast cells in the presence or absence of DPSC℗ in the inflammatory state induced by TNF-α. (**B**). Immunofluorescence images were shown after staining for α-SMA and collagen 1 in fibroblast cells in the presence or absence of DPSC℗ in the inflammatory state induced by TNF-α.

**Figure 5 cells-11-02409-f005:**
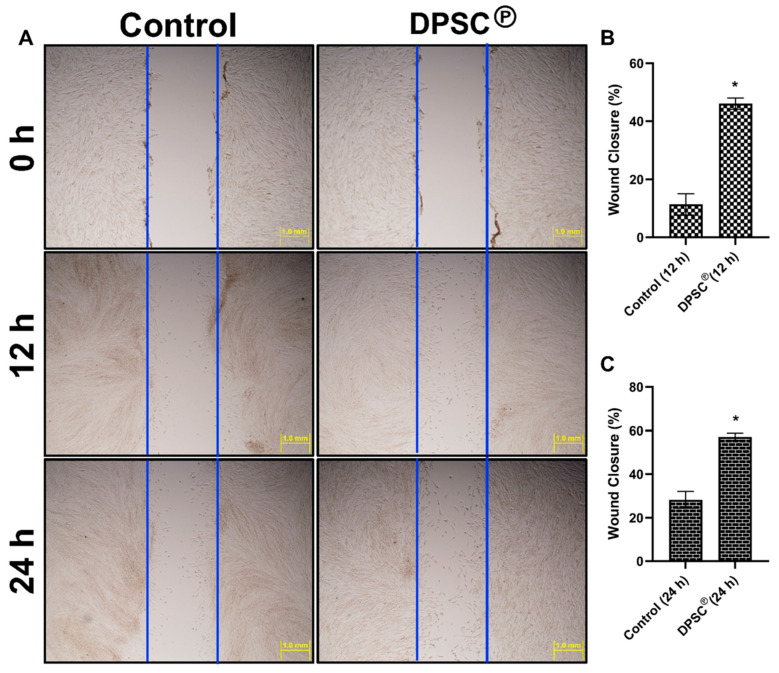
DPSC℗ heals experimental wounds through induction of α-SMA and Collagen 1. (**A**) Experimental scratch wounds healing assay at 12 and 24 h compared with the control assay. Experiments in both conditions were carried out using high-glucose, low-serum (1% FBS) media. Quantified values are shown graphically after 12 h (**B**) and 24 h (**C**). A star (*) represents a significant change (*p* < 0.05) compared to the control group.

**Figure 6 cells-11-02409-f006:**
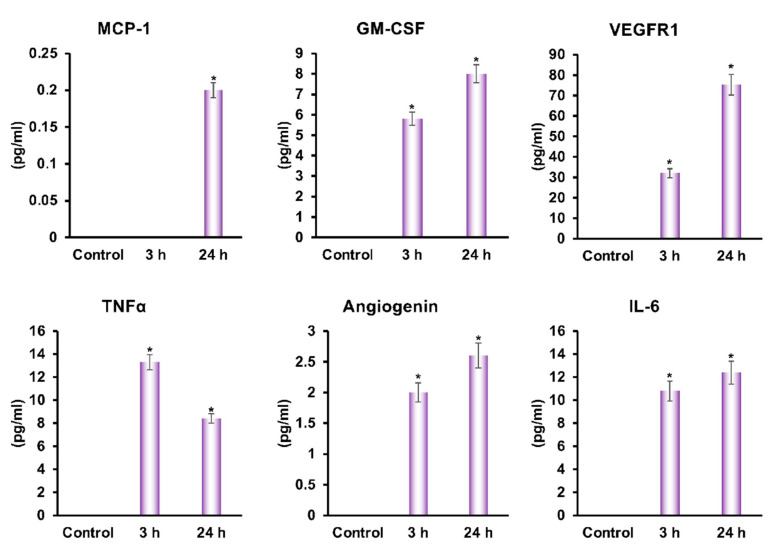
Evaluation of cytokines, angiogenic factors, and growth factors secreted by DPSC that are related to wound healing. The ELISA data are shown graphically for monocyte chemoattractant protein-1 (MCP-1), granulocyte-macrophage colony-stimulating factor (GM-CSF), and vascular endothelial growth factor receptor1 (VEGFR1), TNF-α, angiogenin, and IL-6, in the medium after 3 and 24 h culture of DPSC. Average values were assembled from two different cultures with quadruplicate evaluations. A star (*) represents statistically significant changes (*p* < 0.05) compared to the control.

**Figure 7 cells-11-02409-f007:**
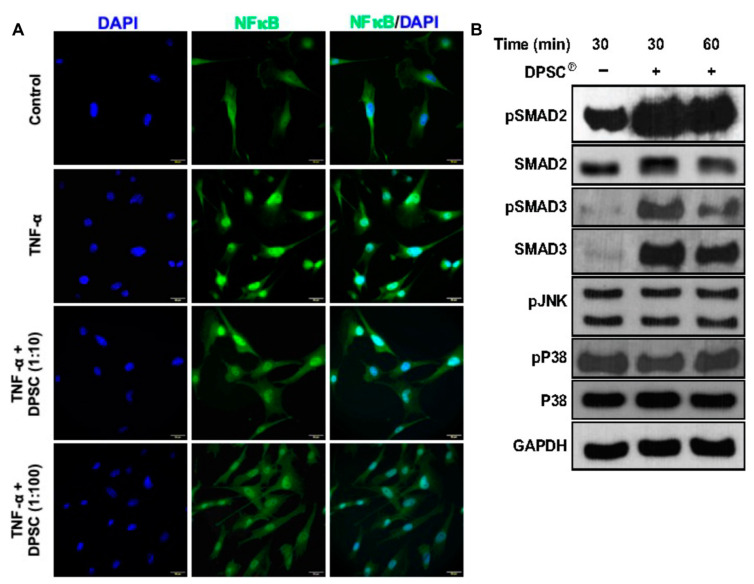
DPSC℗ reduces nuclear translocation of NFkB and induces phosphorylation of SMAD2 and SMAD3. (**A**) Immunofluorescence images are shown for nuclear translocation of NFkB in primary fibroblast cells in the presence or absence of various doses of DPSC in the inflammatory state induced by TNF-α. (**B**) Western blot images of phosphorylation of various signaling pathway molecules are presented, keeping GAPDH as an internal control.

## Data Availability

The datasets used and/or analyzed during the current study will be available on reasonable request.

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
