# Peer review of "DPSC Products Accelerate Wound Healing in Diabetic Mice through Induction of SMAD Molecules"

_cells, 2022, doi:10.3390/cells11152409_

Round 1
Reviewer 1 Report
The manuscript entitled “DPSC products accelerate wound healing in diabetic mice through induction of SMAD molecules” by Carl J Green et al. (2022) aimed to reveal the regenerative effect of dental pulp stem cell (DPSC) derived conditioned media in the diabetic mouse skin wound model. This study contains interesting application of DPSC derived product as candidate for regenerative medicine. However, there are missing context in the manuscript and lack of some details for experiment. Therefore, this manuscript needs to be modified before publication in Cells.
Major comments
1. What is the DPSC product? In the context of manuscript, it looks like authors are indicating some kind of conditioned media obtained after culturing DPSC. Clarify definition of DPSC product.
2. How the DPSC product was prepared in this study? In the material and method section, there’s no mention about the process.
3. In Figure 1B, which condition was used for the Control group? Is it normal media without culturing the DPSC? Or is it another buffer? Make it clear.
4. In Figure 2A, what are the FB and CG sign in the histology image? How can fibroblast be identified in the H&E staining without immunohistochemistry?
5. In Figure 2 and Section 3.2, without quantification, authors cannot say there’s significant difference. Add quantification of FB, CG, and myeloperoxidase.
5. In Figure 4, 5, and 7, the experimental groups are inconsistent. Redesign or re-arrange the experimental groups.
6. In Figure 5, add TNF-a group in the scratch assay.
7. In the Figure 6, what is the meaning of upregulated VEGFR1 secretion? The VEGFR1 is membrane protein (it is kind of receptor protein) and I cannot understand the secretion of VEGFR1.
Author Response
- What is the DPSC product? In the context of manuscript, it looks like authors are indicating some kind of conditioned media obtained after culturing DPSC. Clarify definition of DPSC product.
Answer: We have clarified that the DPSC product, which consists the secretory products of DPSCs that were cultured in 1x PBS overnight before collecting the products that is now clarified in the section 2.5.
- How the DPSC product was prepared in this study? In the material and method section, there’s no mention about the process.
Answer: We have now explained in detail in the Materials and Methods section regarding the definition of DPSC product.
- In Figure 1B, which condition was used for the Control group? Is it normal media without culturing the DPSC? Or is it another buffer? Make it clear.
Answer: We have now provided the clarification about the control group that was vehicle treated control, and was added to the Sections 2.5 and 3.1 in the revised manuscript.
- In Figure 2A, what are the FB and CG sign in the histology image? How can fibroblast be identified in the H&E staining without immunohistochemistry?
Answer: We added abbreviations collagen (CG), that was stained specifically, and have removed references to fibroblasts (FB) and CG in the IHC images, as further confirmation is needed to specifically identify fibroblasts and collagen secretory cells in those H&E images.
- In Figure 2 and Section 3.2, without quantification, authors cannot say there’s significant difference. Add quantification of FB, CG, and myeloperoxidase.
Answer: We corrected the wording by changing from significantly to noticeably as this is qualitative data.
- In Figure 4, 5, and 7, the experimental groups are inconsistent. Redesign or re-arrange the experimental groups. It is also unclear how long TNF-α and DPSC supernatant were added to the fibroblasts before RT-PCR, staining, or WB.
Answer: Figure 5D was moved to become figure 4B to better demonstrate the thinking behind the design for each experiment. In vitro wound assays were carried out in low-serum, high glucose media, representing a diabetic condition in both the control group and the DPSC-treated group in Figure 5.
- In Figure 5, add TNF-a group in the scratch assay.
Answer: Figure 5’s scratch wound data was carried out using low-serum, high glucose media (DMEM high glucose (4.5 g/L, 25mM) to represent the diabetic condition in vitro. For this reason, TNF-a was not added to the cultures. To clarify the principles of design behind the experiments, Figure 5D has been moved to Figure 4B, as it better complements the RT-PCR data in Figure 4.
Reviewer 2 Report
I don’t think there is sufficient in vivo data to support the manuscript claims. Quantification is required on cellularization, re-epithelialisation and collagen deposition, and inflammatory cells recruitment. Only some images are not enough.
The authors must provide detail on ELISA kit. The concentration of some detected proteins is under 6 picogram which is not reliable in most of ELISA kits in the market. Most kits have the lowest detection of 16 picogram.
Some in vivo data is needed to show the effect of detected proteins in wound healing.
How did you choose the factors to screen in the secretome? A high-throughput screening is required for the secretome characterisation
Asma shows no elevation in gene expression but high in protein. Why?
Authors have mentioned many molecules in the introduction part but have not done any experiment on them. Please try to Keep it focused on your work.
What do FB and CG mean on H and E images?
Have you done any characterisation test to identify DPSC? If yes, please provide it.
There is no information about the method of collection of DPSC product for in vivo studies. Serum amount, timing, concentration of total protein?
There is no figure 7 c that has been mentioned in discussion.
Author Response
- In the Figure 6, what is the meaning of upregulated VEGFR1 secretion? The VEGFR1 is membrane protein (it is kind of receptor protein) and I cannot understand the secretion of VEGFR1.
Answer: Barleon et al. (PMID: 11806246) demonstrates that VEGFR1 can be a secreted in a soluble form and that is likely what was measured by the ELISA.
- I don’t think there is sufficient in vivodata to support the manuscript claims. Quantification is required on cellularization, re-epithelialisation and collagen deposition, and inflammatory cells recruitment. Only some images are not enough.
Answer: We have quantified data on myeloperoxidase (Neutrophils) and collagen deposition staining as those were stained specifically. As other data on cellularization, re-epithelialization is mostly relative study, and based on H & E staining, we could not quantify them as software captures specific color only and that can skew the result. Number of images were captured at least five from each section. For the sake of space, we are using only limited representative images.
- The authors must provide detail on ELISA kit. The concentration of some detected proteins is under 6 picogram which is not reliable in most of ELISA kits in the market. Most kits have the lowest detection of 16 picogram.
Answer: We respect the reviewer’s interest in the exact ELISA procedure; however, Section 2.7 clearly states that the multiplex ELISA was performed through a third party, Ray Biotech, and that no kits of our own were used for this experiment. Exact protocol used by the third party was not provided, as those are their trade secrets
- Some in vivo data is needed to show the effect of detected proteins in wound healing.
Answer: It is unclear if the reviewer is asking about the DPSC secreted proteins effect wound healing or if the reviewer is asking for the effect of proteins in the therapeutic target e.g. collagen.
- How did you choose the factors to screen in the secretome? A high-throughput screening is required for the secretome characterisation.
Answer: This was a screening of Multiplex ELISA performed by Ray Biotech, Atlanta, USA. We chose only few proteins from the screening that were most relevant to the current study.
- Asma shows no elevation in gene expression but high in protein. Why?
Answer: We did not pursue how this increase occurs, but we hypothesize that is might be due to a decreased post transcriptional modification (such as ubiquitination) of α-SMA.
- Authors have mentioned many molecules in the introduction part but have not done any experiment on them. Please try to Keep it focused on your work.
Answer: Excess information regarding MMPs and TIMPs has been removed from the revised manuscript.
- What do FB and CG mean on H and E images?
Answer: We added abbreviations for collagen (CG).
- Have you done any characterization test to identify DPSC? If yes, please provide it.
Answer: We have routinely characterized the phenotypes of these cells, and published previously, and the data and experimental design is cited in section 2.2 of the revised manuscript. Here is the example of the phenotype.
- There is no information about the method of collection of DPSC product for in vivo studies. Serum amount, timing, concentration of total protein?
Answer: In section 2.5 of the revised manuscript, we clarified that the DPSC product consists of secretory products from DPSCs cultured in 1x PBS overnight before collection of the product.
- There is no figure 7 c that has been mentioned in discussion.
Answer: This was a typo and has been corrected to Figure 7B.
Reviewer 3 Report
Manuscript ID cells-1806696
Title: DPSC products accelerate wound healing in diabetic mice through induction of SMAD molecules.
In this study the authors have investigated the cellular and molecular mechanisms by which DPSC products accelerated diabetic wound closure.
In my opinion the present work is very interesting.
However: 1) the authors could clarify where their idea of using DPSCs for diabetic ulcers starts and 2) the authors could better discuss the aim and the experimental design of the work, information is missing. I could suggest the authors to insert a graphical abstract to better clarify the experimental design presented.
Minor concerns
Section 2.3 line 122 page 3 the formula for scratch assay Repair area percentage as [(Area t0 − Area t/Area t0) × 100]
Section 2.4 is not clear, could the authors explain the procedure indicating more details?
Section 2.7 multiplex ELISA, was a multiplex assay done? If so, which markers were analyzed? all together in the same plate?
2.9 quantitative PCR real time, how were the results expressed?
To note, in the Figure 5 A the pictures are blurry. Moreover, it is not very reliable to evaluate wound healing over time by taking random images, a time lapse experiment should be carried out.
Discussion section line 395 page12, Neutrophils Neutrophils…..repeated two times
The discussion section is long.
The authors could add the number of the experiments and of the replicates considered for the study.
In addition to the markers of inflammation, do the authors have any other evidence on the efficacy of stem cells on the repair of ulcers caused by diabetes?
Given these shortcomings the manuscript requires major revisions.
Author Response
- However: 1) the authors could clarify where their idea of using DPSCs for diabetic ulcers starts and 2) the authors could better discuss the aim and the experimental design of the work, information is missing. I could suggest the authors to insert a graphical abstract to better clarify the experimental design presented.
Answer: We refined the introduction to focus more on the products relevant to the study. This should help put focus on the rationale for the use of DPSCs in the study. A graphical abstract has now been included to more clearly illustrate the design of the study.
- Section 2.3, line 122, page 3 the formula for scratch assay Repair area percentage as [(Area t0 − Area t/Area t0) × 100]
Answer: We edited the formula to reflect this comment and defined the terms of the equation.
- Section 2.4 is not clear, could the authors explain the procedure indicating more details?
Answer: The number of injections was clarified in our methods section.
- Section 2.7 multiplex ELISA, was a multiplex assay done? If so, which markers were analyzed? all together in the same plate?
Answer: Section 2.7 states that the ELISA was done through a third party, Ray Biotech. This was a high-throughput screening, and we chose proteins from the screening that were relevant to the study.
- 9 quantitative PCR real time, how were the results expressed?
Answer: Results were expressed as fold change, 2-∆∆CT, compared to control, and we have now clarified this in the paper.
- To note, in the Figure 5 A the pictures are blurry. Moreover, it is not very reliable to evaluate wound healing over time by taking random images, a time lapse experiment should be carried out.
Answer: Following your suggestion, we have corrected the Figure 5A.
- Discussion section line 395 page12, Neutrophils Neutrophils…..repeated two times
Answer: We have corrected this in the revised version of the article.
- The discussion section is long.
Answer: We respect the reviewer’s desire for a concise manuscript; however, we believe that this discussion is comparable to other published literature, and we believe that all the information is relevant and important for the quality of the paper.
- The authors could add the number of the experiments and of the replicates considered for the study.
Answer: All in vitro was run in triplicate and this clarification was added to Section 2.13. We also added the number of mice used in each group of the wound healing experiment to Sections 2.4 and 2.5
- In addition to the markers of inflammation, do the authors have any other evidence on the efficacy of stem cells on the repair of ulcers caused by diabetes?
Answer: The data in Figure 4 shows increased fibronectin and Col1A1 gene expression in the fibroblasts cells when exposed to DPSC supernatant. Figure 5D (moved to Figure 4B in the revised manuscript) shows increased protein expression of α-SMA in fibroblasts exposed to DPSC supernatant. This at the very least implies another mechanism of action by which DPSC supernatant can induce wound healing in the diabetic condition.
Round 2
Reviewer 1 Report
I think the revised manuscript is acceptable.
Author Response
We have replied to all questions during our previous submission.
Reviewer 2 Report
The manuscript is surely improved following the first round of comments. However, I still think that this research needs more in vivo data to support its claims about inflammation and wound healing. For example, in vitro data shows changes in a-SMA, IL-6 and IL-10 but none of these factors has been investigated in vivo. Only neutrophil quantification at one time-point is not sufficient.
For me, the level of mentioned proteins in the DPSC product is not reliable considering the lack of information on the kit brand and sensitivity.
I am still wondering why the authors were not able to quantify the re-epithelialization and cellularisation as it is a simple analysis using basic light microscopy and the Image J software.
All the best
Author Response

(The authors gave the same response as above.)

Reviewer 3 Report
the new version of the manuscript has been improved
Author Response

(The authors gave the same response as above.)
